# Effects of a Midwife-Coordinated Maternity Care Intervention (ChroPreg) vs. Standard Care in Pregnant Women with Chronic Medical Conditions: Results from a Randomized Controlled Trial

**DOI:** 10.3390/ijerph18157875

**Published:** 2021-07-25

**Authors:** Mie G. de Wolff, Julie Midtgaard, Marianne Johansen, Ane L. Rom, Susanne Rosthøj, Ann Tabor, Hanne K. Hegaard

**Affiliations:** 1Department of Obstetrics, Copenhagen University Hospital, Rigshospitalet, 2100 Copenhagen, Denmark; Marianne.Johansen@regionh.dk (M.J.); Ane.Lilleoere.Rom@regionh.dk (A.L.R.); hanne.kristine.hegaard@regionh.dk (H.K.H.); 2The Research Unit for Women’s and Children’s Health, The Juliane Marie Centre, Copenhagen University Hospital, Rigshospitalet, 2100 Copenhagen, Denmark; 3Department of Clinical Medicine, University of Copenhagen, 2200 Copenhagen, Denmark; julie.midtgaard.klausen@regionh.dk; 4Mental Health Centre Glostrup, University of Copenhagen, 2600 Glostrup, Denmark; 5Unit for Pregnancy and Heart Disease, Copenhagen University Hospital, Rigshospitalet, 2100 Copenhagen, Denmark; 6Research Unit of Gynecology and Obstetrics, Department of Clinical Research, University of Southern Denmark, 5230 Odense, Denmark; 7Section of Biostatistics, Department of Public Health, University of Copenhagen, 1014 Copenhagen, Denmark; sr@biostat.ku.dk; 8Center of Fetal Medicine and Pregnancy, Department of Obstetrics, Copenhagen University Hospital, Rigshospitalet, 2100 Copenhagen, Denmark; Ann.Tabor@regionh.dk

**Keywords:** maternity care, midwife-led intervention, satisfaction, randomized controlled trial, antenatal care, chronic medical conditions, integrated care

## Abstract

The proportion of childbearing women with pre-existing chronic medical conditions (CMC) is rising. In a randomized controlled trial, we aimed to evaluate the effects of a midwife-coordinated maternity care intervention (ChroPreg) in pregnant women with CMC. The intervention consisted of three main components: (1) Midwife-coordinated and individualized care, (2) Additional ante-and postpartum consultations, and (3) Specialized known midwives. The primary outcome was the total length of hospital stay (LOS). Secondary outcomes were patient-reported outcomes measuring psychological well-being and satisfaction with maternity care, health utilization, and maternal and infant outcomes. A total of 362 women were randomized to the ChroPreg intervention (*n* = 131) or Standard Care (*n* = 131). No differences in LOS were found between groups (median 3.0 days, ChroPreg group 0.1% lower LOS, 95% CI −7.8 to 7%, *p* = 0.97). Women in the ChroPreg group reported being more satisfied with maternity care measured by the Pregnancy and Childbirth Questionnaire (PCQ) compared with the Standard Care group (mean PCQ 104.5 vs. 98.2, mean difference 6.3, 95% CI 3.0–10.0, *p* < 0.0001). In conclusion, the ChroPreg intervention did not reduce LOS. However, women in the ChroPreg group were more satisfied with maternity care.

## 1. Introduction

Worldwide the population of pregnant women living with chronic medical conditions (CMC) is increasing [1,2,3], and in Denmark, the current prevalence is estimated to be 16–21% of all childbearing women [2,4]. The reason for this rise in CMCs among pregnant women is multifactorial [1,2]. Still, an enhanced registration of diagnoses and awareness of specific conditions, better treatment options (including medications), fertility treatment, and a general increase in maternal age, and obesity rates contribute to the overall rise in pregnancies among women with CMC [2,5]. The most frequently reported chronic conditions in the previous studies were diseases of the thyroid gland, asthma and allergic diseases, depression, hypertensive disorders, and migraine [1,2,6]. Less frequently reported conditions were epilepsy, cardiac disease, and diabetes mellitus [1,2,6]. Pregnancy affected by CMC is associated with an increased risk of adverse outcomes such as miscarriage, pre-eclampsia, preterm birth, operative delivery, congenital malformations, and hospitalization compared to women without CMC (1,4–6). Women with CMC also risk deterioration of their CMC during pregnancy or postpartum [7,8,9]. Finally, a higher proportion of women with CMC suffer from perinatal depression, anxiety, and elevated worries than women without CMC [10,11]. Despite the implementation of multidisciplinary maternity care (obstetrician-led care involving midwives and medical specialists) [12,13,14,15], a positive effect on maternal and neonatal outcomes remains to be established [14,15].

Qualitative evidence suggests that women with CMC also experience pregnancy and childbirth as fragmented with insufficient attention given to the women’s individual needs and the normal aspects of the childbearing experience [16].

Evidence from previous randomized controlled trials (RCTs) including pregnant women with obstetric risk factors suggests that midwifery continuity of care models [17,18,19], care-coordination [20,21], implementation of specialized multidisciplinary teams [22], and antenatal and postpartum telephone support [23,24] have the potential to improve the quality of maternity care with positive effects on patient-reported outcomes, health utilization and maternal and infant outcomes. However, no prior trials have explicitly focused on maternity care for women with CMCs, and the question of how to deliver the best maternity care to women with CMC remains essentially unexplored. Therefore, this study aimed to evaluate the effects of a midwife-coordinated maternity care intervention (ChroPreg) delivered to pregnant women with pre-existing CMC. Specifically, we hypothesized that a midwife-coordinated, woman-centered, and specialized intervention, in addition to Standard Care, would shorten LOS in comparison with Standard Care alone. Moreover, we expected the intervention to be superior to Standard Care in relation to psychological well-being and satisfaction with maternity care.

## 2. Materials and Methods

### 2.1. Study Design

This trial used a two-arm parallel-group randomized trial design (1:1). The methods applied and a detailed description of the intervention have been published previously [25]. The reporting of the trial follows the CONSORT guidelines [26]. No changes were made to the intervention or prespecified outcomes outlined in the original study protocol [25]. The trial was registered at ClinicalTrials.gov on 27 April 2018 (NCT03511508).

### 2.2. Setting and Participants

The trial was conducted at the Department of Obstetrics, Rigshospitalet, Copenhagen University Hospital from October 2018–August 2020, with follow-up completion by October 2020. The Department is a tertiary referral center with ≥5000 births annually. Pregnant women with a single fetus, one or more CMCs before pregnancy, aged ≥ 18 years, and who understood written and spoken Danish were eligible to participate. We defined CMC as any prolonged medical condition diagnosed >6 months before pregnancy, with continued reoccurrence and a need for medical treatment [27,28]. An overview of the included main categories of CMC can be seen in the Results section. Exclusion criteria were substance abuse disorders, diabetes type 1 or 2, cardiac conditions, or mental illness unless combined with other CMCs, as these women were referred to already existing multidisciplinary care teams. We chose to exclude women with these CMCs because the care they would receive in these teams, if allocated to standard care, would differ significantly from the standard care for the remaining CMCs in the trial. Eligible women were offered verbal and written information. If consenting to receive further information, they were contacted by phone after the 12th pregnancy week by a research midwife for in-depth information. If the women agreed to participate, a written consent form was signed, and a baseline questionnaire was filled in before randomization. 

### 2.3. Randomization and Blinding

Randomization was performed in a 1:1 ratio between the ChroPreg intervention and Standard Care. The randomly generated allocation sequence was computer-generated with concealed varying block sizes. Allocation to study group was centrally administered by the online clinical trial management and randomization software EasyTrial (Easytrial.net, Aalborg, Denmark). 

### 2.4. Standard Care

In Denmark, maternity care is tax-financed and free of charge, and most women give birth at public hospitals (97%) [29]. Antenatal care is delivered in a primary care-based collaboration between hospital-based midwives and the general practitioner. If pregnancy-related complications arise, women are referred to hospital-based obstetricians for clinical evaluation. In Denmark, midwives are authorized to independently provide care for low-risk pregnant and laboring women [30]. Routinely, women with CMC receive collaborative care provided by obstetricians, medical specialists, and midwives in a tertiary hospital [12]. 

Women in the Standard Care group were referred to an obstetrician. An individual care plan was designed dependent on the character and severity of the respective CMC and the pregnancy-associated risks. Participants were also referred to a midwife for standard care antenatal consultations scheduled around weeks 14–18, 28, 35, 38, and 40. Consultations include physical examination, discussion about lifestyle and physical and mental well-being, symptoms of normal and complicated pregnancies, and breastfeeding-related issues. Antenatal classes consisted of two group sessions in an auditorium (90 min) delivered by midwives. Intrapartum care was provided by the midwives on call on the labor ward. Postnatal care was delivered by midwives, nurses, and obstetricians on the maternity ward. According to individual needs, postpartum consultations could be scheduled after hospital discharge at the woman’s request. This could be a debriefing session with a midwife, or, for women who had experienced severe or unexpected complications, a postnatal consultation with an obstetrician was offered.

### 2.5. ChroPreg Intervention in Addition to Standard Care

The design of the ChroPreg intervention was based on a literature review of maternity care interventions aiming to improve the quality of maternity care for women with high-risk pregnancies and aiming to fit the individual care needs of women with CMC [19,21,22,23,31]. The three main components of the ChroPreg intervention were: (1) Midwife-coordinated and individualized care, (2) *additional ante-and postpartum consultations*, and (3) *specialized known midwives* (Figure 1). What differentiated the ChroPreg intervention from Standard Care was that the specialized midwives, who undertook all antenatal and postpartum midwife consultations, had the role of care coordinators between all health care providers involved in providing maternity care for the women. During each visit, the specialized midwife followed up on appointments with, e.g., obstetricians and other medical specialists to help the woman understand her care plan and assist her in integrating information given during all consultations to assure that any questions or uncertainties would be addressed. When necessary, the midwife would coordinate additional communication or consultations with involved care providers. In addition to the routine visits, two additional (1-h long) visits were scheduled antenatally. Postpartum follow-up in the weeks after birth and a face-to-face postpartum debriefing session to facilitate evaluation and processing of the childbearing and birth experience were also planned [32]. To ensure flexibility and individualization of care, unlimited access to e-mail consultations and weekly telephone hours were available (Appendix A Appendix A and the study protocol [25]).

### 2.6. Prespecified Outcomes

#### 2.6.1. Primary Outcome

The primary outcome was LOS, defined as the total number of whole days of hospitalization from study inclusion until two weeks after birth. Data for this outcome were collected from electronic medical records. LOS is widely used as an objective health utilization outcome in health care research [33,34]. In this study, a potential reduction in LOS encompassed two critical issues: the patient perspective with a reduced need for hospitalization and the health economic aspect with a possible reduction of costs [34].

#### 2.6.2. Secondary Outcomes

All patient-reported outcomes were collected through self-administered electronic questionnaires during the third trimester (33–37 weeks) and two months after birth. Psychological well-being was assessed by the World Health Organization Well-being Index (WHO-5) [35], the Edinburgh Postnatal Depression Scale (EPDS) [36], the Cambridge Worry Scale (CWS) [37], and the Short Form-12 Health Survey (SF-12) [38]. Satisfaction with maternity care was measured by The Pregnancy and Childbirth Questionnaire (PCQ) [39], consisting of a total scale score and two subscales measuring satisfaction with maternity care during pregnancy and delivery, respectively. The Pregnancy sub-scale is further divided into two domains; Personal treatment, and Information and education. The Delivery sub-scale measures the Personal treatment domain during delivery. 

Secondary healthcare utilization outcomes and maternal and infant outcomes were collected from electronic medical records and described in detail in the study protocol [25]. 

### 2.7. Statistical Analysis

Data are presented by median values with Inter Quartile Ranges (IQR) and counts with percentages as appropriate. The power calculation was described in the study protocol [25]. Expecting a drop-out rate of 5% and allocating 129 women to each group, the power was 80%, in order to detect a reduction in LOS of 25%. LOS in the two groups was compared using a t-test on the log scale to allow for a right-skewed distribution. As prespecified, we also performed the comparison adjusted for age and parity. Further, a chi-square test was used to compare the proportion of women with LOS ≤ 3 days in each group.

The secondary patient-reported outcomes (WHO-5, EPDS, CWS, SF-12) were compared at 33–37 weeks of gestation and two months postpartum. The comparisons were performed using a covariance pattern model, including the baseline measurement as an outcome. The models included the interaction between allocation group and time (baseline/33–37 weeks/two months postpartum) with the constraint that the means at baseline are equal due to randomization [40]. An unstructured covariance pattern was used to model the correlation between the three measurements for each woman. The mean PCQ and CWS scores were compared between the allocation groups using the Welch t-test. Maternal and infant outcomes and secondary health utilization outcomes were compared using the Chi-square, Fisher, or Wilcoxon test as appropriate. The analysis of the mode of delivery was supplemented by a logistic regression analysis adjusting for parity. Calculations were performed using R version 4.0.2.

### 2.8. Changes Due to COVID-19

In response to the global Covid-19 Pandemic [41], from 13 March 2019–8 June 2020, a national lock-down was issued in Denmark [42], affecting certain areas of maternity care, including the ChroPreg intervention. All ChroPreg consultations were retained and converted from face-to-face to telephone consultations. At routine antenatal visits, women were not allowed to bring their partners. During birth and postpartum, women could bring only one person for support. These changes affected both groups equally. 

## 3. Results

### 3.1. Study Population

From 12 October 2018–22 January 2020, all women referred for antenatal care at Rigshospitalet were screened for eligibility (*n* = 6608). Of these, 357 women were invited to participate. After verbal and written information, 63 women declined to participate. Between invitation and randomization, 32 women were excluded. In total, 262 women were randomized to either the ChroPreg intervention (*n* = 131) or Standard Care (*n* = 131). In each group, one woman withdrew consent after randomization. Moreover, the screening process had failed to identify that two women had pre-existing type 2 diabetes and they were therefore excluded. Thus, 258 women were included in the Intention-to-treat (ITT) analysis (Figure 2). 

The study population was primarily well-educated, employed, non-smoking, cohabiting, with a normal BMI and a planned pregnancy. In the ChroPreg group, 50.8% were nulliparous compared with 59.4% in the control group. A total of 18.5% of all had conceived by assisted reproductive technologies (ART). Most women had one CMC, and the most prevalent types were endocrinological conditions, neurological conditions, and rheumatological conditions (Table 1).

Women in the ChroPreg intervention had a median of six antenatal consultations with their midwife, 90% participated in the postpartum telephone follow-up, and 85% in the debriefing session. The optional intervention components of weekly telephone hours and e-mail consultation were utilized by 27% and 33% of the ChroPreg participants, respectively (Table 2). 

### 3.2. Primary Outcome

We found no differences in LOS between the two allocation groups. The median LOS during the study period was three days in both groups (Table 3). Women in the ChroPreg group had 0.1% lower LOS than women in the Standard Care group (95% CI −7.8 to 7%, *p* = 0.97). When adjusting for age and parity, women in the ChroPreg group had 1.8% higher LOS (95%CI −5.0–9.1%) than the Standard Care group. In the ChroPreg group, 84/130 (65%) women had LOS ≤ 3 days compared with 75/128 (59%) in the Standard Care group (*p* = 0.32). 

### 3.3. Secondary Outcomes

The overall response rate to the online questionnaires was 96.5% in the 3rd trimester and 94.6% two months postpartum. We found no differences in any patient-reported outcomes measuring psychological well-being or health-related quality of life (Table 4). An increased level of satisfaction with maternity care measured by the PCQ was found in the ChroPreg group compared with the standard group; mean total PCQ score 104.5 vs. 98.2 (MD difference 6.3 points 95% CI 3.0–10.0, *p* < 0.0001). A higher level of satisfaction was also found concerning the two domains of the pregnancy subscale; women in the ChroPreg group scored 3.3 points higher in Personal treatment (48.8 points vs. 45.5, mean diff. 3.3, 95% CI 2.0–5.0, *p* < 0.0001) and 2.0 points higher in Information/education (25.8 points vs. 23.8 points, CI 1.0–3.0, *p* = 0.003). We found no difference in the delivery subscale.

We found no differences in maternal and infant outcomes between groups (Table 5). 

A non-significant difference was found in the mode of birth, where 98 women in the ChroPreg group had a vaginal birth compared with 84 women in the Standard Care group (75% vs. 66%, *p* = 0.10). Due to a higher proportion of multipara in the ChroPreg group, an adjusted analysis was performed, and parity could not explain this tendency (*p* = 0.11). 

Women in the ChroPreg group had a total median of 11 (IQR 9–13) planned consultations with midwives and obstetric doctors vs. 9 (IQR 8–12) in the Standard Care group (Table 6). Women in the ChroPreg group had a median of 4 (IQR 3–5) planned telephone consultations vs. 2 (IQR 1–4) in the Standard Care group. No differences between the groups were detected regarding unscheduled consultations. During the study period, two women had a late termination of pregnancy (week 22–24) on fetal indication, and one woman experienced a stillbirth (week 30). These cases were found to be unrelated to the intervention. 

## 4. Discussion

### 4.1. Main Results

In this randomized controlled trial, we found no evidence to support that a midwifery-coordinated maternity care intervention could reduce the total LOS for childbearing women with CMC. Women in the ChroPreg group were, however, more satisfied with the maternity care they had received. Women in the ChroPreg group were especially satisfied with the care received during pregnancy in relation to the two domains, Personal treatment and Information, and education. The intervention was found to be safe, and adherence was high.

### 4.2. Interpretation of Results

Worldwide, the average LOS during pregnancy and childbirth has gradually declined [43]. In 2001 the average LOS for all Danish pregnant women was 4.0 days, and in 2018 it had decreased to 2.7 days [44]. Women with CMC are, however, expected to have longer LOS [45,46]. We based our sample size calculation on local data with an average LOS of 3.9 days in the target population. To show a significant reduction in LOS, we considered one day of reduction in LOS (25%) to be clinically significant [25]. During the study period, women in both groups reduced the median LOS by 0.9 days compared to the data initially used to calculate the sample size. Therefore, it may not have been possible or even advisable to reduce LOS further, as a further reduction might potentially negatively impact neonatal readmissions [33], breastfeeding rates, and patient experience [47]. A substantial part of the explanation for the overall reduction of LOS may be ascribed to recent local organizational changes at the study site (December 2019), where all women with uncomplicated deliveries were routinely discharged shortly after birth for outpatient postpartum care [48]. This included 29% of the women in this trial (data not shown) who previously would have been offered a routine postpartum stay. 

In line with other studies on midwife-led maternity care interventions [18,19,49,50], we found an increased level of satisfaction with maternity care among women who received the ChroPreg intervention. Maternal satisfaction is an important outcome when measuring patient experience [51,52]. It is widely recognized as one of three essential pillars in the quality of healthcare and clinical effectiveness, and patient safety [53]. A review of studies evaluating maternal satisfaction with maternity care found that the primary component of satisfaction was a positive relational experience with the primary healthcare provider [54], allowing for an intimate and trusting relationship in which women felt involved in decision-making in antenatal care. This is in line with our findings from the pregnancy domain of the PCQ Personal treatment, which contains items related to the relationship with the midwife and perceived involvement in the decision-making process, rated higher in the intervention group. In the previous studies [18,19,49,50], satisfaction was measured in midwifery continuity of care models, where known midwives provide continuity of care throughout the childbearing experience from antenatal intrapartum and postpartum care [18,19,55] for women at both high and low risk. The ChroPreg intervention included only continuity of care provided by a known specialized midwife throughout all antenatal and postnatal consultations but not intrapartum, which may explain why we found no difference in satisfaction with care during birth. However, in line with our findings, a recent qualitative study from the UK evaluated the implementation of midwifery continuity of care for women of all risks throughout pregnancy and postpartum that did not include intrapartum care and found that women were highly satisfied with the relational continuity, the extra time and flexibility in consultation compared with women receiving Standard Care [56]. 

### 4.3. Strengths and Limitations

Recruitment of participants for the current trial progressed according to plan, and eligible women with CMC were interested in participating (81%). Adherence to all intervention elements was overall high, indicating that the women found the intervention meaningful and relevant.

We had access to complete data on the primary outcome and all other outcomes collected from electronic medical records, ensuring a high internal validity. High response rates were also obtained for self-reported data collected through electronic questionnaires. 

We used validated psychometric tools to measure the patient-reported outcomes [57]. The PCQ has solid psychometric properties [39]. As well as measuring overall satisfaction with maternity care, it distinguishes between antenatal and delivery care [58], allowing for in-depth exploration of elements of maternal satisfaction, which has previously been requested in a Cochrane review evaluating the effect of midwife-led models of care [17]. However, a limitation of using the PCQ is that it does not assess women’s satisfaction postpartum, which was also an intervention component. Further, the PCQ does not provide cut-off values of satisfaction (high or low levels) but measures satisfaction as a qualitative continuum [39]. Therefore, the results do not suggest that women in the Standard Care group were unsatisfied—the results merely imply that women in the ChroPreg group were more satisfied with the care they received. 

Another strength of the intervention is the potential reduction in the need to transfer pregnant women to other care teams. The experience from this trial showed that only one woman in the ChroPreg group (vs. 15 in the Standard Care group) needed a referral to other specialist midwives. This is in line with findings from a previous review of interventions of integrated maternity care [31].

However, some limitations of the study design and methods should be discussed. First, we chose LOS as our primary outcome for this trial. Based on studies in mixed populations and populations with women of high risk that found a reduction of LOS in midwife-led interventions, we wished to assess if an intervention given to a population of women with CMC would reduce overall LOS. We hypothesized that, through improved self-reported psychological well-being and health-related QoL, women would feel empowered and more ready to cope with everyday life in pregnancy and the immediate postpartum period [25]. However, we found no difference in LOS or self-reported psychological well-being or health-related QoL, and our hypothesis was rejected. We may have overlooked the clear association between medical and pregnancy-related problems as the primary causes of hospitalization in the study period, and we do not have any reason to believe that the intervention could change the pathophysiology of any of the CMCs or their interaction with pregnancy.

Secondly, we excluded women with cardiac conditions, diabetes mellitus, substance abuse disorders, and psychiatric conditions as only CMC, because these women had already received maternity care which included some of the elements included in the ChroPreg intervention. Therefore, standard care would differ between these women and women with other CMCs. These conditions are, however, associated with risk of severe adverse maternal and neonatal outcomes [13,59], and therefore the inclusion of these women would have been desirable when evaluating the effect of the intervention on LOS as well as secondary maternal and neonatal outcomes. 

We included a wide variety of CMCs. According to national recommendations, to be eligible for the trial the pregnant woman should have at least one CMC that needed tertiary obstetrician-led care [12]. By including CMCs such as asthma and thyroid disease, which are not as strongly associated with adverse maternal and neonatal outcomes as, e.g., hypertensive disorders and epilepsy [7,13], there is a risk that this may have affected the results in the direction of no differences between groups in any secondary maternal and infant outcomes.

Third, we were not able to test a full scale midwife continuity of care intervention that also included intrapartum care. At the recruiting hospital it was not an organizational option of care at the time of the trial, but future studies should assess the effect of a midwife continuity of care model for women with CMC.

Generalizing the findings from this trial to other populations of childbearing women with CMC must be done cautiously. It is necessary to consider that the study population was well-educated, averagely weighted, and non-smoking. Therefore, the results may not be generalizable to pregnant women with less advantageous characteristics. Blinding of participants and specialized midwives was not possible and introduced a risk of bias. We can, therefore, not preclude a degree of researcher or social desirability bias. We aimed to reduce this risk of bias by measuring outcomes with validated instruments and introducing an objectively measurable primary endpoint. 

For this trial, it was not possible to conduct a full-scale economic evaluation of the cost-effectiveness of the intervention [60,61], as we did not have access to data on total health costs. This analysis might have added further to the complete picture of the effects of the intervention and should be included in future studies. The detected difference of a median of two additional visits with midwives and obstetricians during the study period corresponds to the two additional planned antenatal visits, which were an essential component of the intervention, and therefore expected. However, adding two additional visits to the maternity care model for women with CMC is not without financial costs and may hinder the more widespread implementation of the intervention elements in clinical settings. However, interventions to improve the patient experience of maternity care towards a positive experience of the childbearing process are highlighted by the WHO as in important aim for maternity care and should be considered when building future models of maternity care [51].

## 5. Conclusions

Childbearing women with CMC are considered to have high-risk pregnancies, and maternity care for this population requires a high level of flexibility and specialization to ensure optimal outcomes. To our knowledge, this RCT was the first to compare a midwife-coordinated maternity care intervention with Standard Care for women with CMC. We found no difference in the LOS between the groups. We did, however, find that women in the ChroPreg group were overall more satisfied with maternity care. The intervention was feasible, safe, and easy to implement in a clinical setting. Further studies among women with CMC should consider measures of satisfaction with postpartum care and explore the effects of continuity of midwife-led care throughout the childbearing experience, including intrapartum care.

## Figures and Tables

**Figure 1 ijerph-18-07875-f001:**
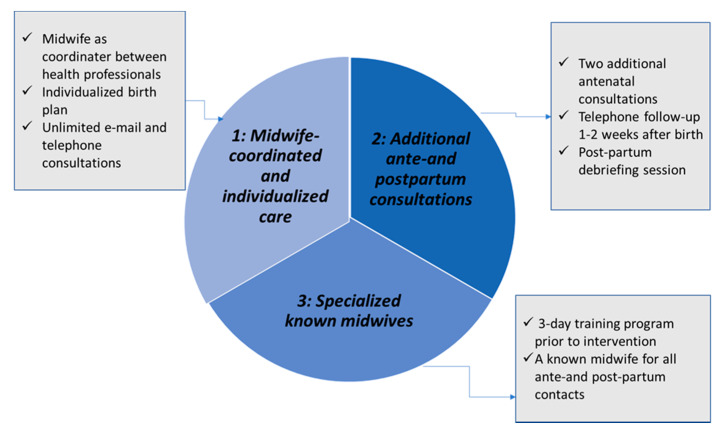
Main components of the ChroPreg intervention.

**Figure 2 ijerph-18-07875-f002:**
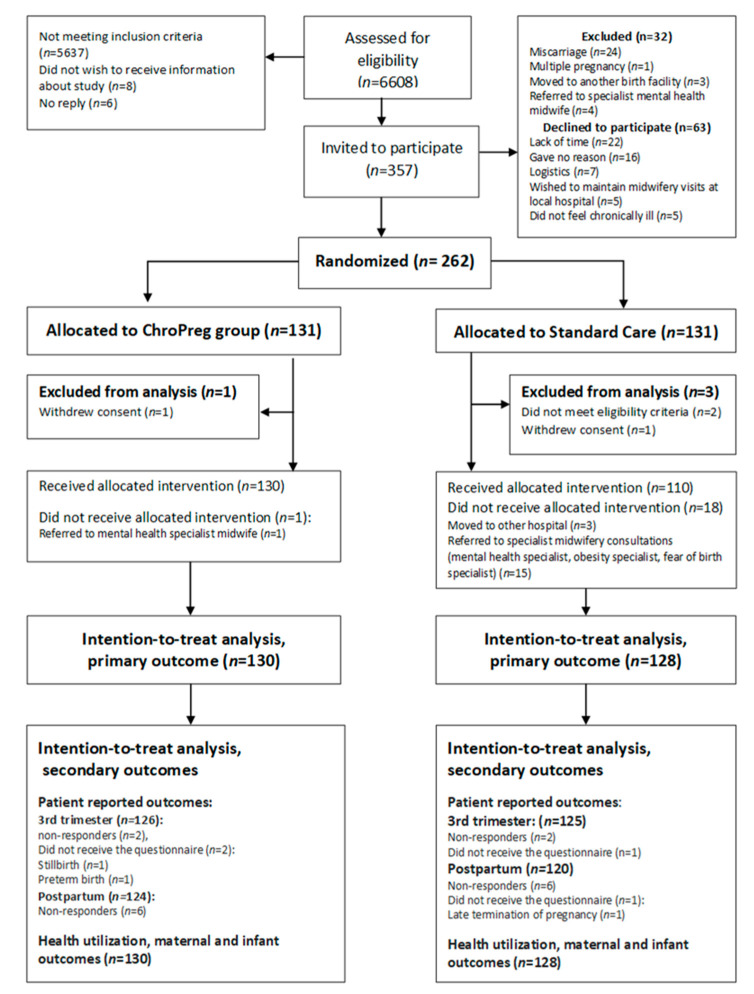
Flowchart of the study population.

**Table 1 ijerph-18-07875-t001:** Self-reported baseline characteristics of the study population (*n* = 258).

	ChroPreg Group (*n* = 130)	Standard Care Group (*n* = 128)
	*n* (%)	*n* (%)
Maternal Age (years)		
<25	2 (1.5)	4 (3.1)
25–29	25 (19.2)	32 (25.0)
30–34	64 (49.2)	48 (37.5)
35–39	29 (22.3)	32 (25.0)
≥40	10 (7.7)	12 (9.4)
Mean (SD)	33.2 (4.2)	33.2 (4.9)
Parity		
Nulliparous	66 (50.8)	76 (59.4)
Multiparous	64 (49.2)	52 (40.6)
Mode of conception		
Spontaneous	106 (81.5)	104 (81.2)
ART	24 (18.5)	24 (18.8)
Smoking before pregnancy		
Yes	16 (12.3)	15 (11.7)
No	114 (87.7)	113 (88.3)
Smoking now		
Yes	3 (2.3)	0 (0.0)
No	127 (97.7)	128 (100)
Cohabitation		
Yes	124 (95.4)	118 (92.2)
No	6 (4.6)	10 (7.8)
Body Mass Index (kg/m^2^)		
<18.5	5 (3.8)	8 (6.3)
18.5–24.9	95 (73.1)	95 (74.2)
25–29.9	16 (12.3)	17 (13.3)
≥30	14 (10.8)	8 (6.3)
Mean (SD)	23.9 (4.7)	22.7 (3.9)
Education		
Compulsory	9 (6.9)	8 (6.3)
Skilled	4 (3.1)	5 (3.9)
Tertiary education (1–2 years)	9 (6.9)	8 (6.3)
Bachelor or equivalent (3–4 years)	36 (27.7)	43 (33.5)
Master or equivalent	72 (55.4)	64 (50.0)
Occupation		
Employed	101 (77.7)	103 (80.5)
Unemployed	6 (4.6)	8 (6.3)
Student	15 (11.5)	12 (9.4)
Other *	8 (6.2)	5 (3.9)
Medication before pregnancy		
No medication	29 (22.3)	25 (19.5)
1–2 medications	68 (52.3)	76 (59.4)
3–5 medications	33 (25.4)	27 (21.1)
Current medication		
No medication	43 (33.1)	40 (31.3)
1–2 medications	64 (49.2)	63 (49.2)
3–5 medications	23 (17.7)	25 (19.5)
Degree of pregnancy planning		
Highly planned	70 (53.8)	71 (55.5)
Fairly planned	24 (18.5)	24 (18.8)
Neither planned nor unplanned	22 (16.9)	25 (19.5)
Fairly unplanned	6 (4.6)	4 (3.1)
Highly unplanned	8 (6.2)	4 (3.1)
Number of CMCs		
One	99 (76.2)	92 (71.9)
Two	20 (15.4)	29 (22.7)
Three	10 (7.7)	6 (4.7)
Four	1 (0.8)	1 (0.8)
Types of CMC:		
Endocrinological disease ^a^	34 (19.7)	32 (18.6)
Neurological disease ^b^	41 (23.7)	34 (19.8)
Rheumatological disease ^c^	30 (17.3)	36 (20.9)
Hematological disease ^d^	10 (5.8)	15 (8.7)
Bowel disease ^e^	25 (14.4)	18 (10.5)
Hypertension	9 (5.2)	8 (4.7)
Lung disease ^f^	8 (4.6)	8 (4.7)
Kidney disease	3 (1.7)	7 (4.1)
Liver disease	1 (0.6)	1 (0.6)
Psychiatric disease	1 (0.6)	1 (0.6)
Endometriosis	2 (1.2)	5 (2.9)
Other CMCs **	9 (5.2)	7 (4.1)

^a^ E.g., hypothyroidism, hyperthyroidism; ^b^ E.g., epilepsy and multiple sclerosis; ^c^ E.g., rheumatoid arthritis, systematic lupus erythematosus; ^d^ E.g., Factor five Leiden, hemophilia A, Protein S/C deficiency; ^e^ E.g., ulcerative colitis, Morbus Crohn’s disease, inflammatory bowel disease; ^f^ E.g., asthma (only included if need for medical treatment during pregnancy), cystic fibrosis. Abbreviations: ART, assisted reproductive technologies; CMC, Chronic medical condition. * Including stay at home mothers, maternity leave, pension e.g., ** Other CMCs include; injuries, congenital malformations, skin conditions, and malignancies.

**Table 2 ijerph-18-07875-t002:** Adherence to the ChroPreg intervention (*n* = 130).

Component in Intervention	
Antenatal midwife consultations (median/range)	6 (2–10)
Postpartum follow-up by telephone (*n*/%) *	117 (90)
Postpartum debriefing session (*n*/%) *	110 (85)
E-mail consultation ** (*n*/%)	43 (33)
Weekly telephone hours ** (*n*/%)	35 (27)

* The ChroPreg midwives attempted to contact all participants by telephone at least twice. ** Optional components of the intervention.

**Table 3 ijerph-18-07875-t003:** Length of hospital stay (LOS) in days according to group allocation.

Allocation	*n*	Median	IQR	Range	LOS ≤ 3 Days
					*n* (%)
ChroPreg group	130	3	2–4.75	1–35	84 (65)
Standard Care group	128	3	2–4.0	1–23	75 (59)

**Table 4 ijerph-18-07875-t004:** Patient-reported outcomes for the intention-to-treat population according to group allocation. All analyses are based on the constrained covariance pattern model.

		ChroPreg Group	Standard Care Group			
**WHO Five-Item Well-Being Index**	***n***	**Mean**	**Mean**	**Mean Difference**	**95% CI**	***p*-Value**
Total scale score 33–37 weeks	250	57.6	58.3	−0.7	−4.2–2.8	0.70
(missing CPG = 4, SCG = 4)						
Total scale score Two months postpartum	240	67.5	65.5	2.0	−1.6–5.6	0.27
(missing CPG = 8, SCG = 10)						
**Edinburgh Postnatal Depression Scale (EPDS)**	***n***	***n* (%)**	***n* (%)**			***p*-Value**
EPDS ≥ 10 33–37 weeks	249	27 (21)	21 (17)			0.38
(missing = 4, SCG = 5)						
EPDS ≥ 10 Two months postpartum	239	25 (21)	19 (16)			0.36
(missing = IG = 9, SCG = 10)						
**Cambridge Worry Scale**	***n***	**Mean**	**Mean**	**Mean Difference**	**95% CI**	***p*-Value**
Total scale score 33–37 weeks	250	16.5	15.52	1.0	−0.8–2.8	0.27
(missing CPG = 4, SCG = 4)						
**SF-12**	***n***	**Mean**	**Mean**	**Mean Difference**	**95% CI**	***p*-Value**
Physical Component Summary week 33–37	250	38.9	40.2	−1.3	−3.5–0.8	0.23
(missing = CPG = 4, SCG = 4)						
Physical Component Summary Two months postpartum	242	49.3	47.7	−0.5	−2.7–1.7	0.66
(missing = CPG = 7, SCG = 9)						
Mental component summary week 33–37	250	51.1	51.4	−0.3	−2.4–1.8	0.76
(missing = CPG = 4, SCG = 4)						
Mental component summary Two months postpartum	242	49.1	49.7	−0.6	−2.8–1.5	0.55
(missing = CPG = 7, SCG = 9)						
**Pregnancy and Childbirth Questionnaire**	***n***	**Mean**	**Mean**	**Mean Difference**	**95% CI**	***p*-Value**
Total scale score 8 weeks postpartum	241	104.5	98.2	6.3	3.0–10.0	0.001
(missing CPG = 8, SCG = 9)						
Pregnancy subscale—Personal treatment Two months postpartum	241	48.8	45.5	3.3	2.0–5.0	<0.0001
(missing CPG = 8, SCG = 9)						
Pregnancy subscale—Information and education Two months postpartum (missing CPG = 8, SCG = 9)	241	25.8	23.8	2.0	1.0–3.0	0.003
Delivery subscale–Personal treatment Two months postpartum (missing CPG = 8, SCG = 11)	239	29.9	29.4	0.5	−1.0–2.0	0.51

Abbreviations: CPG; ChroPreg group, SCG; Standard Care group.

**Table 5 ijerph-18-07875-t005:** Maternal and infant outcomes for the intention-to-treat population according to allocation.

	ChroPreg Group		Standard Care Group		
	***n***	**%**	***n***	**%**	***p*** **-Value**
Pregnancy complications ^a^	19	15	20	16	0.82
Antenatal outpatient telemonitoring	3	2	4	3	0.72 **
Intention to breastfeed (yes) (Assessed 33–37 weeks)	121	96	118	95	0.73
Labor onset ^b^					0.60
Spontaneous	67	60	58	56	
Induced	45	40	45	44	
Mode of birth					
Vaginal	98	75	84	66	0.10 *
Cesarean section	32	25	44	34	
Preterm delivery	11	8	8	6	0.60
Use of epidural analgesia	52	40	44	34.4	0.35
Apgar Score ≤ 7 at 5 min	1	1	1	1	N/A
Breastfeeding (yes) (Assessed 8 weeks postpartum)	106	87	98	83	0.41
	**Median**	**IQR**	**Median**	**IQR**	***p*** **-Value ***
Gestational age at birth (days)	276	269–284	278	268–283	0.84
Birth weight (kilograms)	3.3	3.1–3.8	3.4	3.0–3.7	0.35
Intended length of breastfeeding (months)	9	6–12	8	6–12	0.16
(Assessed eight weeks postpartum)					

^a^ Preeclampsia, hypertensive disorders of pregnancy, gestational diabetes mellitus. ^b^ In this analysis, only women with a trial of vaginal birth are included (*n* = 215). * Wilcoxon. ** Fischer’s exact test.

**Table 6 ijerph-18-07875-t006:** Health utilization for the intention-to-treat population according to allocation.

	ChroPreg Group		Standard Care Group		
	Median	IQR	Median	IQR	*p*-Value *
Number of planned visits with a midwife Antenatally	6	5–7	4	3.25–5	<0.0001
Number of planned visits with an obstetrician Antenatally	3	1.25–4	3	2–5	0.21
Total number of planned visits with midwife or obstetrician whole study period **	11	9–13	9	8–12	0.0004
Total number of unscheduled visits whole study period	1	0–2	1	0–2	0.63
Total number of planned telephone consultations whole study period	4	3–5	2	1–4	<0.0001
Total number of unscheduled telephone consultation whole study period	2	1–3	2	1–3	0.69

* Wilcoxon ** consultations with midwives, obstetricians, outpatient visits for labor induction, postpartum follow-up.

## Data Availability

Data available on request due to restrictions, e.g., privacy or ethicalThe data presented in this study are available on request from the corresponding author. The data are not publicly available due to restrictions due to the risk of identifying participants with unique characteristics.

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
