# Peer review of "Effects of a Midwife-Coordinated Maternity Care Intervention (ChroPreg) vs. Standard Care in Pregnant Women with Chronic Medical Conditions: Results from a Randomized Controlled Trial"

_ijerph, 2021, doi:10.3390/ijerph18157875_

Round 1
Reviewer 1 Report
It has been a pleasure to review this manuscript.
I think there is no doubt that midwives are the personnel specialized in the care of the pregnant woman both in the prenatal period, as well as during childbirth and the puerperium.
I believe that in order to offer quality care to women it is necessary for the personnel who attend them to form a multidisciplinary team, composed of course by obstetricians, midwives, physiotherapists, psychologists ...
Of course, I believe that all maternity wards must be coordinated by midwives to ensure optimal results.
Overall, the design of this research looks pretty good to me.
In the introductory section, I think it will be important to define the chronic diseases that can affect pregnant women.
It is true that both the material and method section and Table 1 are defined, but I think the introduction is a bit incomplete without this information.
On the other hand, the analytical study seemed correct to me, the results are clearly stated and the conclusions support the results.
Thanks
Regards
Reviewer 2 Report
Overall, this is a reasonable RCT evaluating midwifery co-ordinated obstetric care. The study design has some issues, namely, no continuity during intrapartum or inpatient care included in the study protocol, exclusion of some cardiac and endocrine disorders, and a choice of a primary outcome that has limited clinical significance.
The negative and positive findings from the study are in line with previous reports that suggest this model of care does not improve clinical outcomes for mothers and their babies, but it does improve patient satisfaction. However, the negative important negative findings are useful to report.
A better study design might have improved the study and influenced the results.
Abstract
The conclusion sentence mentions safety and ease of implementation, but the study did not assess safety nor service burden and cost. This should be amended.
Introduction
Worth mentioning what are the most common chronic medical conditions encountered in maternity (? Hypertension, diabetes, cardiovascular diseases). Are many of these conditions due to changing demographics in the patient groups such as age, obesity?
What is meant by highly specialized maternity care? Many national guidelines recommend multidisciplinary and co-ordinated care but a positive effect on maternal and neonatal outcomes has not been established.
Taylor, C., McCance, D.R., Chappell, L. et al. Implementation of guidelines for multidisciplinary team management of pregnancy in women with pre-existing diabetes or cardiac conditions: results from a UK national survey. BMC Pregnancy Childbirth 17, 434 (2017). https://doi.org/10.1186/s12884-017-1609-9
Bick, D., Beake, S., Chappell, L. et al. Management of pregnant and postnatal women with pre-existing diabetes or cardiac disease using multi-disciplinary team models of care: a systematic review. BMC Pregnancy Childbirth 14, 428 (2014). https://doi.org/10.1186/s12884-014-0428-5
My impression of continuity of care models is that there is no evidence for a statistically significant improvement in maternal and neonatal outcomes reported thus far. Does this study aim to address this lack of knowledge?
Methods
Please can the authors specify the chronic medical conditions included in the RCT, please list or reference the table in the results? My understanding is that the excluded CMCs (substance abuse disorders, diabetes type 1 or 2, cardiac conditions, or mental illness) have a significant affect on pregnancy pathology and outcomes. Does the current standard of care model at the recruiting site bear a resemblance to COC? Were they excluded because the authors assumed current practise would influence their results? Please explain.
The intervention appears to be the use of a specific team of midwives to co-ordinate care between professionals. This is already provided in hospitals in other countries as part of their standard care. How is their intervention different? Was their continuity of care intervention providing antenatal, intrapartum, inpatient care and postnatal periods? If not, why not?
Specified outcomes
The study was aimed at evaluating the effects of a midwife-coordinated maternity care intervention (ChroPreg) delivered to pregnant women with pre-existing CMC.
Why did the authors choose length of hospital stay as the primary outcome? Readers of the manuscript may find this is very strange choice.
COC intervention would not prevent patients from developing medical disorders, but it might allow earlier detection, and provision of outpatient care. Was the intension to evaluate indirectly outpatient care? Or better/faster access to healthcare professionals though the co-ordinating midwife? If so, please can the authors describe how these are linked to LOS? Most readers would expect the study to evaluate improvements in maternal and neonatal outcomes, because these have not been reported to improve with COC models in the literature. What is the clinical improvement here with a reduced LOS as a primary outcome and how was this measured and delivered?
Improvements in patient reported outcomes have been published elsewhere.
The secondary maternal and infant outcomes should be specified/listed (you can refer to the table in the results).
Statistics
Why did the authors present the data as medians and IQR but then transform the data for LOS when a non-parametric analysis would suffice?
Results
The CMCs are very broad. Can the authors specify the most common endocrine, neurological, rheumatological and bowel disorders? Disorders like hypothyroidism with adequate thyroxine replacement are unlikely to see significant benefits from the study intervention. Could this be why there was limited improvements between cohorts?
LOS was increased with the study intervention and there was no difference in secondary maternal and infant outcomes. Did the authors consider if the stay in hospital was directly associated with the medical disorder or other unrelated pregnancy issues? What about the number of hospital admissions for the medical problem? I.e linking LOS with clinical benefit.
The additional appointments in the study group may have a cost and service burden. How might this impact widespread implementation?
Discussion
This is well written and considers many of the above points.
This section should also include bias surrounding exclusion of certain medical conditions, and the use of a segmented continuity of care model lacking continuity during intrapartum and inpatient care.
Also, can the authors discuss how many of the chronic medical conditions included in the study groups required specialist care because they would otherwise have had a significant impact on maternal and neonatal outcomes? Perhaps some of the included disorders with limited clinical consequences for mothers and their babies would see little difference with their COC model versus standard care.
As it stands this study demonstrates that the intervention is welcomed by patients, but it does not provide a clinical benefit and is likely to have a greater cost.
